# High Effectiveness of a 14-Day Concomitant Therapy for *Helicobacter pylori* Treatment in Primary Care. An Observational Multicenter Study

**DOI:** 10.3390/jcm9082410

**Published:** 2020-07-28

**Authors:** Llum Olmedo, Rafael Azagra, Amada Aguyé, Marta Pascual, Xavier Calvet, Emili Gené

**Affiliations:** 1ABS Manresa 3, Althaia, Xarxa Assistencial de Manresa, Programa de Doctorado en Ciencias de la Salud, Universitat Internacional de Catalunya, 08243 Barcelona, Spain; mlolmedo@uic.es; 2Departamento de Medicina, Universitat Internacional de Catalunya, 08195 Sant Cugat, Barcelona, Spain; razagral@telefonica.net (R.A.); mpascual.girona.ics@gencat.cat (M.P.); emgene@uic.es (E.G.); 3USR Metropolitana Nord IDIAP Jordi Gol, Departament de Medicina, Universitat Autònoma de Barcelona, 08017 Barcelona, Spain; aaguye.ics@gencat.cat; 4CAP Badia del Vallés, Institut Català de la Salut, 08214 Badia del Vallés, Barcelona, Spain; 5ABS Granollers, 08401 Granollers, Barcelona, Spain; 6ABS Arbúcies-Sant Hilari, Institut Català de la Salut, 17401 Arbúcies, Spain; 7Unitat de Malalties Digestives, Hospital Universitari Parc Taulí. Institut d’Investigació i Innovació Parc Taulí I3PT, Sabadell, Universitat Autònoma de Barcelona, 08208 Sabadell, Girona, Spain; 8Centro de Investigación Biomédica en Red en Enfermedades Digestivas y Hepáticas (CIBERehd), 028029 Madrid, Spain; 9Servicio de Urgencias, Hospital Universitari Parc Taulí, Institut d’Investigació i Innovació Parc Taulí I3PT. Sabadell, Universitat Autònoma de Barcelona, 08208 Sabadell, Barcelona, Spain

**Keywords:** *Helicobacter*, eradication, treatment, concomitant, primary care, multicenter

## Abstract

Background: The current cure rates with triple therapy combining a proton-pump inhibitor, amoxicillin and clarithromycin are unacceptably low. Aims: To evaluate the efficacy of a 14-day concomitant therapy as an empirical first-line treatment for curing *Helicobacter pylori* (*Hp*) infection in primary care. Methods: Patients from six primary care centers in Catalonia -Spain- were included consecutively. *Hp* status pre and post treatment was assessed according to local clinical practice protocol. A 14-day concomitant therapy (amoxicillin 1 g, clarithromycin 500 mg and metronidazole 500 mg plus omeprazole 20 mg, all drugs administered twice daily) was prescribed. Adherence to therapy and adverse events were assessed by personal interview. Results: 112 patients were enrolled. Mean age was 46.7 ± 16.1 years. Main indication for treatment was non-investigated dyspepsia (83%). *Hp* eradication was achieved in 100 of the 112 patients. Eradication rates were 89.3% (95% CI: 81.7–94.1) by intention-to-treat (ITT) analysis and 91.7% (95% CI; 84.6–95.9) per protocol (PP). No major side effects were reported; 104 (92.8%) patients complete the treatment. Forty-seven patients (42%) complained of mild side effects (metallic taste, nausea). Low adherence to treatment (*p* = 0.004) and significant adverse events (*p* = 0.004) were the variables associated with treatment failure. Conclusions: In primary care, a 14-day concomitant therapy is highly effective and well tolerated.

## 1. Introduction

*Helicobacter pylori* infection causes a wide range of gastric disorders, from dyspepsia and peptic ulcer disease to life-threatening tumors such as gastric carcinoma and mucosa associated lymphoid tissue-lymphoma [1]. Certain extra-digestive diseases such as idiopathic thrombocytopenic purpura and idiopathic iron-deficiency anemia are also associated with *Helicobacter pylori* [2]. The infection causes disease via different mechanisms that provoke a chronic inflammatory response in gastric mucosa. The CagA cytotoxin seems to play a major role, especially in the pathogenesis of cancer [3,4]. Therefore, eradication of *Helicobacter pylori* is an important issue; however, the best treatment is still to be defined.

Since the bacteria is able to survive in various locations (intraluminal, intercellular and intracellular), the treatment needs to achieve adequate therapeutic levels in all these different microenvironments so as to eradicate the infection. Previous approaches have combined an acid suppressor with many different antibiotics [5]. Even with these combinations, the ideal treatment remains elusive. Triple therapy combining a proton-pump inhibitor (PPI), amoxicillin and clarithromycin for 7 to 10 days has been the undisputed choice for first-line therapy worldwide for many years [6,7]. In the last ten years, however, many studies worldwide have found unacceptably low cure rates for this “classical” or “legacy” triple therapy [8,9].

One of the most successful alternatives for first-line treatment was concomitant therapy including a PPI, amoxicillin, metronidazole and clarithromycin for 10 to 14 days. Clinical trials and meta-analysis have shown that this therapy cures over 90% of patients even in areas with moderately high clarithromycin resistances [10,11,12]. On the basis of these results, concomitant therapy is widely recommended worldwide [13,14,15,16,17,18].

However, concomitant therapy has some drawbacks: (a) its recommended schedule (10 or 14 days) is quite long; (b) it requires four different drugs which must be prescribed separately; (c) given its complexity, the schedule must be carefully explained to patients in order to ensure adherence; finally, (d) concomitant therapy includes clarithromycin and metronidazole, two antibiotics with a relatively poor digestive tolerance.

For a long time, the standard treatment in primary care has been triple therapy; indeed, a 2017 survey found that standard triple therapy remained the preferred schedule for 56.4% of primary care physicians [19]. In the specialized setting, the more effective quadruple therapy is already the preferred treatment, but in primary care the combination of the complexity and relatively poor tolerance of quadruple concomitant therapy may well be an important drawback. To our knowledge, all previous studies evaluating concomitant therapy have been performed in specialized settings, and the vast majority were hospital-based [11,12,20,21,22,23]. As *Helicobacter pylori* is treated mainly at primary care level, it seems important to ascertain whether concomitant therapy maintains its effectiveness when used in a primary care under normal clinical practice conditions. The survey mentioned above found that only 27.7% of primary care physicians used a concomitant treatment [19].

In this study, we performed a trial to evaluate the safety, applicability and efficacy of a concomitant schedule combining omeprazole 20 mg, amoxicillin 1 g, metronidazole 500 mg and clarithromycin 500 mg twice per day, given over a 14-day period, as an empirical first-line treatment for curing *Helicobacter pylori* infection in a primary care setting.

## 2. Methods

Patients with indication for *Helicobacter pylori* eradication according to the IV Spanish Consensus Conference, were considered for recruitment in the study. The study was performed at six primary care centers (PCC) in Catalonia.

Exclusion criteria were (a) previous *Helicobacter pylori* treatment; (b) allergy to any of the antibiotics used; (c) pregnancy; (d) age lower than 18 years.

The study was labeled as a prospective follow-up post-authorization study (EPA-SP) by the Department of Medicines for Human Use of the Spanish Agency of Medicines and Sanitary Products. All patients were included consecutively. Written informed consent was obtained from all patients and the ethics committee responsible of the different PCCs approved the study protocol (XCC-AMO-2014–01).

*Helicobacter pylori* diagnosis was made by a non-invasive test (urea breath test or *Helicobacter pylori* stool antigen) or, if the patient required an upper gastrointestinal endoscopy, with a rapid urease test or histology. A single positive test was accepted as evidence of infection for two reasons: (a) it was routine practice and (b) given the high prevalence of *Helicobacter pylori* in our setting, in patients with dyspepsia or ulcer the positive predictive value of a single test for *Hp* infection before treatment is extremely high.

Treatment consisted of a 14-day therapy combining three antibiotics (amoxicillin 1 g, clarithromycin 500 mg and metronidazole 500 mg) plus omeprazole 20 mg all administered twice per day. Patients were encouraged to avoid alcoholic beverages during treatment to prevent the possible side-effects of their interaction with metronidazole.

Cure of *Helicobacter pylori* infection was evaluated in accordance with the routine non-invasive test used by each PCC. In patients with gastric ulcer, needing a second endoscopy to rule out gastric cancer, eradication was assessed by histology. Control test was performed at least eight weeks after completion of treatment. Patients were instructed to avoid PPI for at least two weeks and antibiotics for four weeks before the diagnostic test. Adherence to therapy and adverse events were assessed by personal interview after the end of antibiotic treatment.

## 3. Statistical Analysis

The overall eradication rates and their 95% confidence intervals were obtained by intention to treat (ITT) and per protocol (PP). ITT analysis included all eligible patients who had received at least one dose of treatment, and PP analysis all those who had a valid follow-up test to check the cure of the *Helicobacter pylori* infection. Quantitative variables were given as mean ± S.D. and qualitative variables were presented as percentages. A univariate analysis including age (divided into quartiles), sex, indication for treatment, diagnostic test for *Helicobacter pylori* infection, *Helicobacter pylori* eradication test, adherence and presence of severe adverse events was performed using the chi-squared test or the Mann–Whitney *U*-test. Calculations were performed using the SPSS 21 software.

Sample size calculation: For a reference population of 100,000 people and an expected ITT cure rate of 90%, a sample of 82 patients was necessary to estimate the efficacy of the treatment with a ± 5% error margin and a 95% confidence interval. Assuming a loss during follow-up up to 20%, the sample size required was 98 patients.

## 4. Results

One hundred and twelve patients were included in the study. Forty-five per cent were men and mean age was 46.7 ± 16 years. Demographic and clinical characteristics and the indications for eradication are shown in Table 1. Diagnosis of *Helicobacter pylori* infection previous to treatment was made by stool test (67%), urea breath test (21%), CLO-test (8%) or histology (4.5%).

Main indications for treatment were uninvestigated dyspepsia (83%), functional dyspepsia (12.5%) and peptic ulcer (3.5%).

*Helicobacter pylori* eradication was achieved in 100 of the 109 patients who returned for follow-up. Eradication rates were 91.7% (95% CI: 84.6–95.9) by PP analysis and 89.3% (95% CI: 81.7–94.1) by ITT (Figure 1). *Helicobacter pylori* eradication by intention to treat and per protocol for demographic, clinical characteristics and indications for treatment are shown in Table 2.

The treatment was well tolerated: no severe side effects (requiring either treatment or hospitalization) were reported, and 47 patients (41.9%) presented minor to moderate side effects, the most frequent were metallic taste (*n* = 21), mild diarrhea (*n* = 15), abdominal pain (*n* = 13) and nausea or vomiting (*n* = 13). All side effects disappeared shortly after the end of treatment. One hundred and four patients (92.8%) of the patients reported complete adherence to treatment. In the remaining eight patients (7.2%) the treatment was stopped between three and ten days, all because of digestive intolerance. Four of them were cured after 3, 5, 8 and 10 days; one was not cured after three days and three patients (who took the treatment for 3, 4 and 5 days) empirically received second line therapy without testing. These three patients were considered “not cured” in the analysis (Figure 2).

### Factors Influencing the Efficacy of Therapy

There were no statistical differences in the cure rates regarding indication for treatment, sex, age, PCC, tobacco use and diagnostic test either before or after treatment. Adherence to treatment (*p* = 0.004) and significant side effects (*p* = 0.004) were the only variables associated with treatment failure.

## 5. Discussion

The present study shows that concomitant treatment seems to be as effective in primary care and under clinical practice conditions as in hospital-based clinical trials. The PP cure rate of 91.7% was promising and very similar to those obtained with the same treatment schedule during 14 days in previous Spanish trials: 90.7% [11] and 91% [24]. The therapy achieved adequate cure rates despite the use of low PPI doses, in contrast with other studies in our setting in which high PPI doses were needed to achieve optimal cure rates.

In our trial we did not find differences according to the indication for eradication (ulcer or dyspepsia), ulcer location, sex or age, even though the number of patients included was low. Side effects were mild (diarrhea, metallic taste, nausea or abdominal pain). Most patients, however, completed the treatment despite these mild side effects and the symptoms resolved rapidly after the drugs were withdrawn. Adherence was very good and 92% of patients reported completing the prescribed drugs. The presence of significant adverse events and poor adherence to treatment were significantly related to the treatment failure.

The trial was performed under clinical practice conditions. This is both a strength and a limitation. It is a strength because, as the study conditions did not differ from those of usual clinical practice, the applicability of the results is probably very high. However, it is also a limitation because most patients were diagnosed by a non-invasive test, and pretreatment cultures were not obtained. For this reason, antimicrobial resistance could not be evaluated.

Overall, our results give support to the consensus recommendations that concomitant therapy be used as a first-line treatment in primary care setting too [13,14,15,16,17,18]. Concomitant therapy should become the new gold standard therapy against which new therapies should be compared. It would be very interesting to compare bismuth-containing classical quadruple therapy (PPI, metronidazole, tetracycline and bismuth) with concomitant quadruple therapy in primary care.

In conclusion, 14-day of concomitant therapy combining omeprazole with amoxicillin, clarithromycin and metronidazole twice daily is well tolerated and highly effective in eradicating *Helicobacter pylori* infection in primary care.

## Figures and Tables

**Figure 1 jcm-09-02410-f001:**
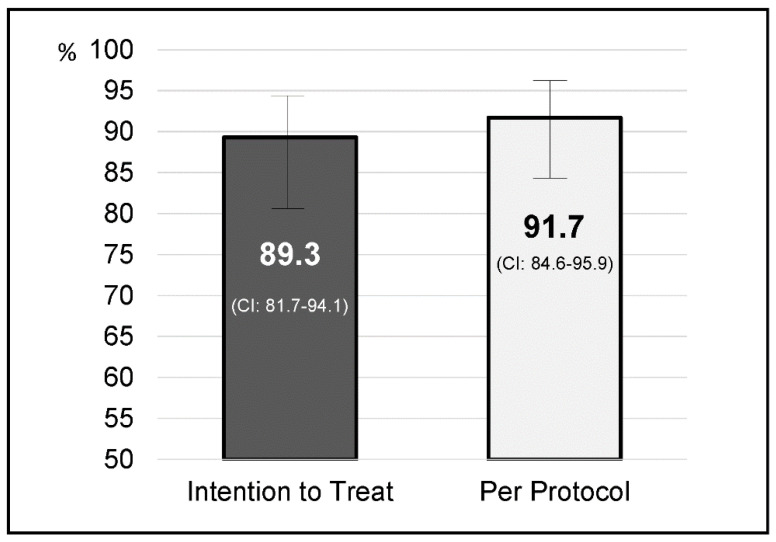
*Helicobacter pylori* eradication by intention to treat and per protocol.

**Figure 2 jcm-09-02410-f002:**
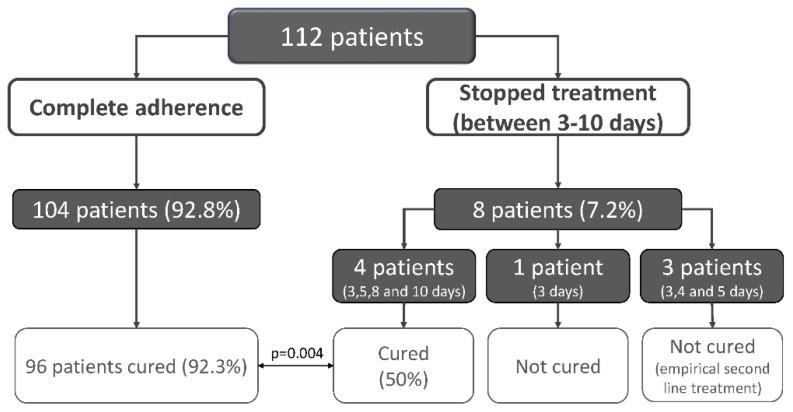
Adherence to treatment and *Helicobacter pylori* eradication.

**Table 1 jcm-09-02410-t001:** Demographic, clinical characteristics and indications for treatment by *Helicobacter pylori* eradication.

**Variables**		**Total**	***p***
Age—Years—(Mean ± SD)		46.7 ± 16.1	0.921
Sex	Men	50 (44.6%)	0.404
	Women	62 (55.4%)	
Tobacco	Smoker	16 (14.3%)	0.404
	Non-smoker	96 (85.7%)	
Primary care centers (PCC)	PCC Bases de Manresa/PCC Barri Antic. Althaia	21 (18.8%)	0.369
	PCC Granollers	54 (48.2%)	
	PCC Badia Valés	8 (7.1%)	
	PCC Arbúcies/PCC St Hilari. Girona	29 (25.9%)	
Main indications	Non-investigated dyspepsia	93 (83%)	0.127
	Functional dyspepsia	14 (12.5%)	
	Peptic ulcer	4 (3.6%)	
	Others	1 (0.9%)	
Diagnostic test previous to treatment	Histology	5 (4.5%)	0.624
	Urea breath test	23 (20.5%)	
	*Helicobacter pylori* stool antigen	75 (67%)	
	Rapid Urease Test	9 (8%)	
Treatment adherence	Complete	104 (92.9%)	0.004
	Partial	8 (7.1%)	
Mild side effects	Yes	47 (42%)	0.004
	No	65 (58%)	
Diagnostic test post treatment	*Helicobacter pylori* stool antigen	96 (85.7%)	0.633
	Urea breath test	16 (14.3%)	
**Variables**		**Total**	***p***
Age—Years—(Mean ± SD)		46.7 ± 16.1	0.921
Sex	Men	50 (44.6%)	0.404
	Women	62 (55.4%)	
Tobacco	Smoker	16 (14.3%)	0.404
	Non-smoker	96 (85.7%)	
Primary care centers (PCC)	PCC Bases de Manresa/PCC Barri Antic. Althaia	21 (18.8%)	0.369
	PCC Granollers	54 (48.2%)	
	PCC Badia Valés	8 (7.1%)	
	PCC Arbúcies/PCC St Hilari. Girona	29 (25.9%)	
Main indications	Non-investigated dyspepsia	93 (83%)	0.127
	Functional dyspepsia	14 (12.5%)	
	Peptic ulcer	4 (3.6%)	
	Others	1 (0.9%)	
Diagnostic test previous to treatment	Histology	5 (4.5%)	0.624
	Urea breath test	23 (20.5%)	
	*Helicobacter pylori* stool antigen	75 (67%)	
	Rapid Urease Test	9 (8%)	
Treatment adherence	Complete	104 (92.9%)	0.004
	Partial	8 (7.1%)	
Mild side effects	Yes	47 (42%)	0.004
	No	65 (58%)	
Diagnostic test post treatment	*Helicobacter pylori* stool antigen	96 (85.7%)	0.633
	Urea breath test	16 (14.3%)	

**Table 2 jcm-09-02410-t002:** *Helicobacter pylori* eradication by intention to treat and per protocol for demographic, clinical characteristics and indications for treatment.

Variables		Intention to Treat (95% CI)	Per Protocol (95% CI)
Sex	Men	92% (80–97.4)	93.9% (82.2–98.4)
	Women	87.1% (75.6–93,4)	90% (78.8–95.9)
Tobacco	Smoker	93.8% (67.8–99.7)	93.8% (67.8–99.7)
	Non-smoker	88.5% (80–93.8)	91.4% (83.3–96)
Primary care centers (PCC)	PCC Bases de Manresa/PCC Barri Antic. Althaia	85.7% (62.6–96.2)	85.7% (62.6–96.2)
	PCC Granollers	92.6% (81.3–97.6)	92.6% (81.3–97.6)
	PCC Badia Valés	100% (59.8–98.8)	100% (59.8–100)
	PCC Arbúcies/PCC St Hilari. Girona	82.8% (63.6–93.5)	92.3% (73.4–98.7)
Main indications	Non-investigated dyspepsia	91.4% (83.3–95.9)	94.4% (83.3–95.9)
	Functional dyspepsia	71,4% (42–90.4)	71,4% (42–90.4)
	Peptic ulcer	100% (39.6–97.6)	100% (39.6–100)
	Others	100% (5.4–89.2)	100% (5.4–100)
Diagnostic test previous to treatment	Histology	80% (29.9–98.9)	80% (29.9–98.9)
	Urea breath test	91.3% (70.5–98.5)	95.5% (75.2–99.8)
	*Helicobacter pylori* stool antigen	88% (80–94)	90.4% (80.1–95.6)
	Rapid Urease Test	100% (62.9–99)	100% (62.9–100)
Diagnostic test post treatment	*Helicobacter pylori* stool antigen	89.7% (81.4–94.7)	92.6% (84.8–96.7)
	Urea breath test	86.7% (58.4–97.7)	86.7% (58.4–97.7)

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
