# Peer review of "High Effectiveness of a 14-Day Concomitant Therapy for Helicobacter pylori Treatment in Primary Care. An Observational Multicenter Study"

_jcm, 2020, doi:10.3390/jcm9082410_

Round 1
Reviewer 1 Report
The manuscript consists of the study of an alternative therapy for the eradication of Hp. The study is well organized and the literature that supports it is adequate.
However, I would like to point out some short comings:
1) Introduction must be enriched, the mechanism of action of Hp and the mechanisms of action of the treatments currently in use must be specified. highlighted the News of the traial.
2) Line 69 go to line with 2.Experimental section
3) In the statistical analysis section it is necessary to specify the statistical test (s) performed (ANOVA, t-Test etc ...)
Author Response
Point-by-point response
Reviewer 1:
Extensive editing of English language and style required
The text has been edited by an experienced native-language scientific editor.
1) Introduction must be enriched, the mechanism of action of Hp and the mechanisms of action of the treatments currently in use must be specified. highlighted the News of the trial.
We have added two paragraphs in the introduction on the pathogenic mechanism of Hp infection (Page 1 lines 42-44 revised version) and a paragraph on the mechanism of action of the treatment. (Page 2 Lines 47-51 revised version)
2) Line 69 go to line with 2. Experimental section
We apologize for this mistake. “2. Experimental section” has been deleted.
3) In the statistical analysis section it is necessary to specify the statistical test (s) performed (ANOVA, t-Test etc ...)
In the statistical analysis section, we already specified “A univariate analysis including age (divided into quartiles), sex, indication for treatment, diagnostic test for Helicobacter pylori infection, Helicobacter pylori eradication test, adherence and presence of severe adverse events was performed using the Chi-square test or the Mann Whitney U-test. Calculations were performed using the SPSS 21 software” (Line 104-105 original version).
Reviewer 2 Report
This study evaluated quadruple therpy for 14 days against Helicobacter pylori at primary care centers. There seem to be several points to be reconsidered.
Major points
- The authors should show PP and ITT results with reference to sex, tabacco, PCC, main indications, previous and post treatment diagnostic tests, which may give the readers additional information on the reliabilty of the results.
- More kind explanation is neccesary on the differnece between the usual regiment in hospitals and that in this study.
Minor points
1. What is "2. Experimental Section"?
Author Response
Point-by-point response
Reviewer 2:
1) The authors should show PP and ITT results with reference to sex, tobacco, PCC, main indications, previous and post treatment diagnostic tests, which may give the readers additional information on the reliability of the results.
We have added these data as Table 2 (Page 6 Lines 256-258 revised version)
2) More kind explanation is necessary on the difference between the usual regiment in hospitals and that in this study.
We have modified the introduction to clarify this point (Page 2 Lines 68-75 revised version)
1. What is "2. Experimental Section"
We apologize for this mistake. “2. Experimental section” has been deleted.

Round 2
Reviewer 2 Report
The authors well responded the reviewer's comments.